# Study of Exercise Capacity and Quality of Life after SARS-CoV-2 Infection among the Elderly

**DOI:** 10.3390/bs13050381

**Published:** 2023-05-05

**Authors:** Diana Vrabie, Beatrice-Aurelia Abalașei

**Affiliations:** Doctoral School in Sports Science and Physical Education, University Alexandru Ioan Cuza of Iasi, 700554 Iasi, Romania

**Keywords:** COVID-19, disease, elderly, quality of life, 6-MWT, WHOQoL-BREF, Romania

## Abstract

COVID-19 significantly impacted the population by affecting physical health; social distancing and isolation influenced psychological health. This may have negative consequences, especially for older people. There is a lack of studies about the association between COVID-19 and exercise capacity among the elderly and improving quality of life after SARS-CoV-2 infection. This study aims to identify the potential sequelae of the COVID-19 disease regarding physical function and quality of life among people over 65 years old. This study recruited a total of 30 participants. A 6-minute walking test, somatic and functional measurements (including weight, height, HR, blood pressure and SpO_2_%) and the World Health Organisation Quality of Life—BREF Questionnaire were used to assess aerobic capacity and quality of life. Experiencing COVID-19 can negatively impact exercise capacity. The results suggest that men may have worse sequelae than women after experiencing COVID-19. The lower values of SpO_2_ in the COVID-19 group during the 6-MWT indicate a reduction in the gas diffusion capacity, which can be attributed to potential lung damage following having contracted the disease. Lockdown periods seem to have had a significant impact on the physical health, relationships and environment of the elderly people included in this study. We can conclude that physical effort may potentially impact exercise capacity and quality of life among post-COVID-19 elderly in a positive way, but further studies are needed to confirm its benefits.

## 1. Introduction

Multiple studies have reported the existence of a post-acute COVID-19 syndrome called long COVID. It can take many forms, from post-intensive care syndrome (PICS) [1] to pulmonary fibrosis secondary to aggressive COVID-19 pneumonia [2].

Long-lasting symptoms often occur without severe acute infections or pre-existing co-morbidities. Many authors have tried to quantify the impact of the persistent symptoms of COVID-19 on physical function, cognitive function, health-related quality of life and participation in social activities. The primary conclusion was that symptoms experienced after COVID-19 can persist for at least two months and often more than 12 months. The most common symptoms reported are fatigue, brain fog, sleep disturbances, dizziness, dyspnoea, memory loss, palpitations, lack of concentration, pain, anxiety, depression and gastrointestinal problems [3,4,5,6].

Immune system disruption triggered by infection could induce psychopathology, with evidence of psychiatric sequelae following previous coronavirus outbreaks. It was concluded that the spread of the SARS-CoV-2 virus implies important psychological manifestations, such as post-traumatic stress, depression and anxiety. All of the above are associated with a low quality of life [7]. In addition to the prevalence of depression in people who experienced the disease, a meta-analysis of twelve studies revealed that the prevalence of depression in the general population during the COVID-19 pandemic was 25%. The most recent survey of the global prevalence of depression was in 2017, with a value of 3.44%. This means that the rate of depression in the general population was even seven times higher during the COVID-19 pandemic [8].

Regarding rehabilitation after disease, Soril et al. [9] concluded that the effectiveness of pulmonary rehabilitation compared to other types of rehabilitation in post-COVID-19 patients is not known. Further comparative studies are necessary to detect the ideal kind of rehabilitation after infection. The studies should also focus on the importance of exercise in preventing infection, not just in rehabilitation after COVID-19. A study from 2021 concluded that the health improvements obtained after four weeks of exercising seemed to persist after 14 weeks of inactivity due to the COVID-19 lockdown and may have prevented severe functional decline and strength loss in institutionalised older adults [10]. There are also countries that, during the pandemic, advised the population to maintain regular physical activity to avoid the risk of sequelae after experiencing the disease [11,12,13].

The research topic addressed in this article represents a subject of global interest, considering the spreading of the virus and the extent of deaths caused by COVID-19. The elderly population is more susceptible, with an increased mortality rate and high chances of developing severe disease forms [14]. This study aims to determine whether there is any impairment of exercise capacity and quality of life level due to COVID-19 among the most susceptible category of people. The motivation behind the topic comes from the need to find solutions to reduce the sequelae and avoid the severe forms of the disease in potential victims of SARS-CoV-2 infection.

**H1.** *The first hypothesis tested by this study is that exercise capacity will be significantly reduced among patients who contracted SARS-CoV-2 infection compared with a relatively homogenous group of non-infected people*.

**H2.** *The second hypothesis claims that increasing the level of physical activity among post-infected people can improve the quality of life*.

## 2. Materials and Methods

The data in this study were collected in February–March 2022. The study was approved by the Faculty of Physical Education and Sports ethics committee (Approval number 101 bis/3 February 2022). Informed consent was obtained from all subjects involved in the study. The study recruited 30 participants from the Saint Parascheva Retirement House and the Saint Joseph Retirement House (Iasi, Romania). These centres accommodate approximately 250 patients, with more than half being immobilised and unable to care for themselves. The inclusion criteria were as follows: age ≥65 years; no contraindications related to physical exertion; no neurological, rheumatological or orthopaedic dysfunctions that would limit the patient’s mobility; no recent myocardial infarction; and no auditory or visual severe dysfunction or mental illnesses. In the first phase, the subjects included in this study were tested weekly with a RT-PCR and then twice a month with rapid antigen tests. They were admitted to this study due to anamnesis and based on the test results registered in their medical history. From the 30 patients selected, 15 participants were never infected with SARS-CoV-2, while 15 had an infection in the last six months before the examination date, detected by performing an RT-PCR test. None of the participants required hospitalisation during the illness and were treated in the host centre. They were isolated for 14 days in individual rooms with permanent health monitoring and treatment depending on the symptoms.

A 6-minute walk test and the World Health Organisation Quality of Life—BREF Questionnaire were used to assess exercise capacity and quality of life. Somatic and functional measurements were performed using an Omron M2 basic blood pressure device, a Tanita UM-076 scale, an Akyta BLS-1102B pulse oximeter and a tape line (including weight, height, HR, blood pressure, and SpO_2_%).

### 2.1. The 6-MWT

The tests for assessing exercise capacity recommended by British Thoracic Society Guidelines are the 6-minute walk test and the Incremental distance walk test [15]. The distance covered (in metres) is the main result of the 6-MWT. It is recommended to be used in series to record any changes in exercise capacity and response to interventions that may alter or improve exercise capacity over time. Peripheral O_2_ saturation, heart rate (during the effort) and blood pressure (before and after performing the 6-MWT) were measured to assess the patient’s respiratory function. The participants were instructed to walk as much as possible for six minutes along a hallway over 15 m delimited by cones in the turnarounds. Subjects were encouraged with standardised statements, such as “You are doing well” or “Keep up the good work.” They were allowed to stop and rest during the test, but were instructed to resume walking as soon as possible. Subjects were asked about the following symptoms: shortness of breath, chest pain, dizziness or leg pain. The distance covered was recorded for each subject, and the values were interpreted according to the purpose of the research. Enright and Sherrill [16] established reference equations, and we used them to predict the 6-MW distance for each individual depending on age, sex, weight and height (for men: 6MWD = (7.57 × height/cm) − (5.02 × age) − (1.76 × weight/kg) − 309 m; for women: 6MWD = (2.11 × height/cm) − (2.29 × weight/kg) − (5.78 × age) + 667 m)).

### 2.2. World Health Organisation Quality of Life—BREF Questionnaire

The WHOQoL-BREF questionnaire is a helpful tool to create a quality-of-life profile. It evaluates four domains: Physical health, Psychological aspects, Social relationships and Environment. The score of items within each field is used to calculate the domain score. The four domain scores obtained denote an individual’s perception of life quality. Domain scores are scaled positively (higher scores indicate a higher quality of life). The questionnaire and the method to calculate the quality of life score can be visualised on the WHOQoL-BREF assessment form [17] (Appendix A).

### 2.3. Intervention Program

All rehabilitation sessions were performed three times a week, lasted 40 min and were carried out individually or in groups of two. Fifteen patients (COVID-19 group) benefited from a rehabilitation program adapted to their capacity, which included aerobic training, flexibility and muscle toning exercises. The sessions began with a warm-up phase and light exercises to adjust the cardiorespiratory and musculoskeletal system to the effort, reduce dyspnoea and increase joint flexibility. This part of the session lasted an average of 7 min. The endurance training was performed on a stationary bicycle where a moderate-intensity effort was interspersed with periods of rest or low-intensity effort. The exercise intensity was monitored and adapted according to the level of perceived exertion, heart rate and oxygen saturation (constantly measured with a pulse oximeter). The emergency conditions to stop the effort were: worsening dyspnoea; onset of dizziness, palpitations, pallor or tachypnoea; SpO_2_ < 88% or a decrease in SpO_2_ > 4% from baseline; and heart rate <60 or >160 beats per minute. Stationary bicycle training lasted 10 min, with a ratio of 1:2 (20 s of moderate intensity effort and 40 s of low intensity or rest) for the first two weeks. Subsequently, after two weeks (weeks 3–10), the aerobic training was conducted for a period of 10 min with a ratio of 1:1 (20 s of moderate intensity and 20 s of low intensity or rest) and the last two weeks (weeks 10–12), 10 min with a 2:1 ratio (40 s moderate intensity and 20 s low intensity or rest). Muscle toning exercises were performed in the training phase to increase upper- and lower-limb muscle strength. The intensity and duration of the workouts were planned and adapted to the specific characteristics of the patients. The initial load was minimum and adjusted to avoid excessive fatigue. The exercises were performed in three sets of 10 to 15 repetitions.

Progression was made by increasing the number of reps, sets or workload (resistance of the elastic band and dumbbell or sandbag weight). During the cooldown phase, stretching [18], flexibility and low-intensity exercises were performed for approximately 5–7 min.

### 2.4. Statistical Analysis

SPSS IBM 22 (Statistical Package for Social Sciences) was used for statistical analysis. Descriptive data are presented as the arithmetic mean and standard deviation (SD).

An independent Samples *t*-test was used to compare the outcomes between the two independent groups regarding age, HR, blood pressure, SpO_2_ and distance covered in the 6-MWT. Cronbach’s alpha and inter-item correlation matrix were calculated to analyse the internal reliability of the questions in the WHOQoL-BREF Questionnaire. We applied the independent samples median test to assess the differences between males and females in each group regarding baseline characteristics. Because the assumption of the normal distribution was not met regarding SpO_2_, a Mann–Whitney U test was used to compare the outcomes of its values.

In the second part of this study, we used a non-parametric test due to the small number of the sample (<30 subjects). The Wilcoxon paired-sample test was used to determine if there is any statistically significant difference between the initial and final assessment regarding the physical characteristics and the scores of the WHOQoL-BREF Questionnaire. Cohen’s d was used to measure the effect’s size regarding improving WHOQoL-BREF Questionnaire domains scores and functional indices between the two assessments.

The Shapiro–Wilk test for normal distribution was performed in SPSS for the dependent variables (age, weight, height, distance covered, SpO_2_, HR) by groups (COVID-19 and non-COVID-19). According to the Z-score (within the range of −1.96–+1.96), we can assume that the data were normally distributed. The data distribution followed the same line on the box plots, except for SpO_2_. The *p*-value of the Shapiro–Wilk test was more significant than 0.05 regarding all variables, except oxygen saturation at rest, which means that, with the exception of SpO_2_, the data had a normal distribution. For testing the homogeneity of variances, we used Levene’s test. A *p*-value greater than 0.05 indicates that the assumption of homogeneity regarding the subjects’ main characteristics (age, weight and height) was met.

## 3. Results

### 3.1. COVID-19 and Exercise Capacity

The participants were divided into two groups, a COVID-19 and non-COVID-19 group. Table 1 features the mean values of the baseline characteristics of the subjects by group (age, sex, weight, height, SpO_2_ and HR at rest) and the standard deviation (SD).

Nine females and six males were included in each group. According to Table 1, the difference between mean age is 4.6 years, with older patients in the COVID-19 group (*p* < 0.05). The weight and height of the subjects were higher in the non-COVID-19 group, on average by 2.45 kg (*p* > 0.05) and 5 cm (*p* > 0.05). The mean values of SpO_2_ at rest were increased in a statistically significant manner among non-infected patients (+0.86%; Mann–Whitney U-test, *p* < 0.05) and the heart rate registered a statistically insignificant difference of means, with 0.47 bpm in favour of the COVID-19 group (*p* > 0.05).

Considering the mean distance covered, it emerges that women infected with SARS-CoV-2 covered a 72.77 m shorter distance than non-infected women. In comparison, infected men covered a 184.16 m shorter distance than non-infected men, as shown in Table 2.

To establish if there are significant differences between males and females in each group, we started by considering that the age, weight and height means were the same across the two sex categories. The independent samples median test outcomes reveal a predominant *p*-value greater than 0.05, as Appendix B shows. Therefore, we can assume that age, weight and HR at rest means are the same across the two sex categories in each particular group.

Regarding the distance covered in the 6-MWT, the *p*-value shows that, in the non-COVID-19 group, the median distance covered was not the same across the sex categories. In contrast to the men in the COVID-19 group, the mean distance covered by non-infected men was more significant than that of women. This suggests that men are more susceptible to being severely affected by COVID-19 than women, which is also explained by the increased number of deaths among men in Romania (23,496 deaths among men and 18,638 deaths among women, reported until October 2021).

To assess whether SARS-CoV-2 infection causes a functional decline after experiencing the disease (with an impairment in aerobic capacity), we compared the distance covered in the 6-MWT within groups using the independent samples *t*-test. As shown in Table 3, people who experienced the disease covered a significantly shorter distance in the 6-MWT than people in the non-COVID-19 group (*p* < 0.05) on average, with 117.33 m.

To appreciate the lung function and peripheral oxygen saturation during effort, we recorded and analysed the oxygen saturation and oscillations every 30 s while performing the 6-MWT. After processing the data, we concluded that, during the 6-MWT, constant changes were recorded in both groups, as shown in Appendix C. Lower SpO_2_ start values were recorded among the infected subjects compared with the non-infected subjects and statistically significant oscillations between groups were recorded during the test. The mean SpO_2_ in post-infected patients was significantly lower at the start of the exercise. SpO_2_ values during the first 90 s after start showed minor, non-significant differences between groups. Two minutes after the starting, the differences in SpO_2_ values between groups were statistically significant, with a higher value in the non-COVID-19 group (*p* < 0.05), except for the difference recorded after 3:30 min from the start.

As shown in Figure 1, SpO_2_ tended to normalise in the non-COVID-19 group towards the end of the effort, while in the COVID-19 group, SpO_2_ continued to decrease.

### 3.2. Physical Activity and Quality of Life

To improve the post-infected patients’ mental and physical health status, we included them (*n* = 15) in a three-month rehabilitation program. The inclusion criteria were as follows: age ≥65 years; no contraindications related to physical exertion; no neurological, rheumatological or orthopaedic dysfunctions that would limit the patient’s mobility; no recent myocardial infarction; and no auditory or visual severe dysfunction or mental illnesses. The program was initiated in April 2022.

During the rehabilitation sessions, peripheral O_2_ saturation, heart rate and perceived exertion on the Borg scale were monitored, and blood pressure was measured at the sessions’ beginning and end. All rehabilitation sessions were performed three times a week (38 sessions), lasted for 40 min, and were carried out individually or in groups of two subjects. The prescription of the exercises was conducted according to the potential of each participant by calculating HR_max_ and depending on self-perceived effort on the Borg scale.

Regarding the quality of life issue, we used the WHOQoL-BREF Questionnaire, which can be found translated into Romanian on the World Health Organisation website [16]. Therefore, we applied it within the COVID-19 group twice: to detect the self-perceived quality of life before and after three months of rehabilitation.

We calculated Cronbach’s alpha and the inter-item correlation to analyse the internal reliability to measure whether individual questionnaire questions provide consistent, appropriate results. The Cronbach’s alpha value was 0.961, meaning that the items in the questionnaire are highly correlated. Table 4 shows the mean value of the inter-item correlations was =0.521, indicating that the items measure the same construct.

After applying the WHOQoL-BREF questionnaire, the following results were registered. The questionnaire field scores at the initial assessment are shown in Table 5. Lockdown periods and social distancing had an increased adverse impact on the quality of life level judging by the Social relationships domain scores. After implementing the rehabilitation program (Appendix D), the mean scores increased by an average of 12.84 points. Considering the Cohen’s d values, it emerged that, between the two assessments, a small effect size was registered regarding social relationships (d < 0.5) and significant effects regarding physical health, mental health and environment (d > 0.8). The differences between the initial and final assessment results are substantial, as shown in Table 5 (*p* < 0.05).

Regarding the baseline characteristics of the subjects, Table 6 shows the differences between the initial and final assessment in terms of exercise capacity and functional indices. The *p*-value suggests that the differences recorded between the mean weight, mean SpO_2_ at rest, HR and distance covered in the 6-MWT are statistically significant between the two assessments. Measuring the effect size with Cohen’s d revealed that mean SpO_2_ and mean distance covered in the 6-MWT registered a large effect size (d > 0.8), while HR at rest showed a small effect size. Both quality of life and exercise capacity among elderly subjects who experienced COVID-19 improved after three months of rehabilitation, superior to the recommended minimal clinically significant difference (MCID) of 30 m for the 6-MWT in chronic lung disease (6-MWT—171.33 m ± 42.57 vs. 218.00 m ± 44.43, MD: 46.66 m, *p* < 0.05).

## 4. Discussion

It is necessary to study more closely the long-term manifestations of COVID-19 to quantify the extent to which physical and psychological aspects are affected and whether spontaneous recovery can occur in different categories of patients. The absence of a control group remains an essential limitation of this study because we cannot assign improvements to the rehabilitation program. We have yet to determine whether spontaneous recovery can occur after COVID-19 and, if so, to what extent. However, the role of physical exercise has been studied and recognised in many pathologies similar to COVID-19 disease in terms of symptoms and long-term manifestations. Many studies showed that frequent clinical manifestations and sequelae regarding pulmonary function and cardiovascular and psychological health could be countered by physical exercise [19,20,21,22]. Because fatigue and decreased exercise tolerance are common clinical symptoms in patients with COVID-19, muscle metabolic function may be affected [23]. There still needs to be a consensus about the optimal strategies to improve exercise tolerance in patients with acute COVID-19 or post-COVID-19 sequelae. Detailed clinical recommendations have been published for physical therapists caring for patients with COVID-19 [24,25], but recommendations for exercise therapy are still suboptimal [26].

A study by Liu et al. (2020) shared the results of six weeks of pulmonary rehabilitation, which assessed patients’ exercise capacity, QoL and mental status. It emerged that, compared to the control group, the mean distance covered in the 6-MWT was significantly longer in the experimental group (212.3 ± 82.5 vs. 157.2 ± 71.7; *p* < 0.05) and was also significant within the experimental group (162.7 ± 72.0, 212.3 ± 82.5) [27]. Regarding the control group, there was no significant difference at six weeks from the baseline (155.7 ± 82.1, 157.2 ± 71.7; *p* > 0.05). A meta-analysis conducted by Chen et al. concluded that the pooled estimate effect of pulmonary rehabilitation in the 6-MWT favoured the experimental group and was superior to the recommended minimal clinically significant difference (MCID) of 30 m for the 6-MWT in chronic lung disease [28,29]. Therefore, physical activity has a good impact on exercise capacity and quality of life among the post-COVID-19 elderly, but further studies are needed to confirm its benefits completely.

Other limitations of this study include the small number of subjects, the difficulty in accessing more elderly care centres, the frailty of this category of people, the associated comorbidities and the mental status. The lack of access to high-performance testing equipment can be considered another research limitation. The limitations of the WHOQoL-BREF Questionnaire concern not evaluating the quality of life, specifically after COVID-19. It was not applied before and after infection, only before and after the rehabilitation program. In future studies, we could evaluate the persistence of quality of life improvements by applying the questionnaire after a more extended period. Following our results, although using different assessment tools, the three studies included in the meta-analysis conducted by Chen et al. concluded that pulmonary rehabilitation could improve QoL for patients who survived COVID-19 [28].

According to the results obtained, it appears that people who contracted COVID-19 covered a significantly shorter distance, on average by 117 m (*p* < 0.05), compared to the people in the non-COVID-19 group, suggesting an impairment in the exercise capacity potentially due to COVID-19 (171.33 ± 42.57 in the COVID-19 group compared with 288.66 ± 76.05 in the non-COVID-19 group). A significant limitation remains the difference in mean age between groups, which can influence the results. That fact partially confirms the hypothesis that the SARS-CoV-2 virus may negatively impact individuals’ exercise capacity. Because of the difference in mean age in this study, further studies with homogenous groups are necessary to elucidate the disease’s actual impact on the body’s functions.

The distance covered in the 6-MWT increased by 46.66 m between assessments, suggesting that the rehabilitation program increased exercise capacity among older adults. This aspect cannot be assumed entirely due to our study’s lack of a control group. However, although with a significant variation in the data, studies in the field of rehabilitation after COVID-19 have shown that respiratory muscle training significantly improved exercise capacity, compared with a control group, regardless of the type of intervention (such as face-to-face or remote, with device-based or not, and with endurance training or not) [27,30,31].

The analysis of distance covered in the 6-MWT between sexes revealed that men in the uninfected group covered a greater distance than women, which did not happen in the infected group. This may suggest that men may have worse sequelae than women after contracting COVID-19, which is also consistent with the pandemic situation reported in Romania (23.496 deaths among men and 18.638 deaths among women, reported until October 2021).

The lower values of SpO_2_ in the COVID-19 group during the 6-MWT can be attributed to the reduction in the gas diffusion capacity due to potential lung damage. A study conducted in 2020 by Klanidhi et al. also concluded that oxygen saturation before the start of the 6-MWT was normal. Still, it decreased significantly after six minutes of walking, possibly due to a decreased respiratory reserve in older people or COVID-19 infection [32]. More high-quality investigations are necessary to confirm this hypothesis.

The WHOQoL-BREF domain scores obtained by the subjects in this study suggest that the disease and isolation measures significantly impacted the physical health and relationships among the older people included in this study.

## 5. Conclusions

The distance covered in the 6-MWT increased by 46.66 m between assessments, suggesting that the rehabilitation program increased exercise capacity among older adults. This aspect cannot be assumed entirely due to our study’s lack of a control group.

Non-infected men covered a greater distance than non-infected women, which did not happen in the COVID-19 group. This may suggest that men may have worse sequelae than women after contracting COVID-19.

The lower values of SpO_2_ in the COVID-19 group during the 6-MWT can be attributed to the reduction in the gas diffusion capacity due to potential lung damage after the disease.

Disease and isolation measures affected more the physical health and relationships of the older people included in this study, suggesting that physical inactivity, lockdown periods and social distancing have an increased negative impact on the quality of life level.

After all, we can conclude that physical effort may potentially impact exercise capacity and quality of life among the post-COVID-19 elderly in a positive way, but further studies are needed to confirm its benefits. For a rapid dissemination of the results, a schematic design of the study can be found in Appendix E.

A future research direction may be a follow-up study performed by conducting pre- and post-infection assessments of the participants. Further research is needed to understand the mechanisms underlying persistent symptoms and the best way to combat them.

## Figures and Tables

**Figure 1 behavsci-13-00381-f001:**
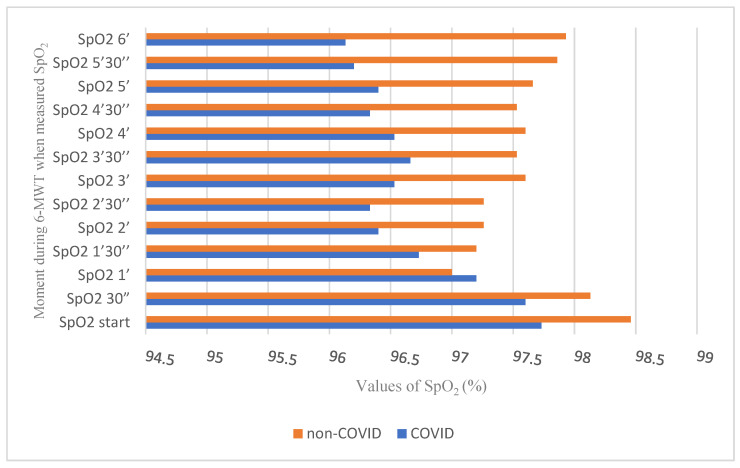
Oscillation of peripheral oxygen saturation values during the 6-MWT in the COVID-19 and non-COVID-19 groups.

**Table 1 behavsci-13-00381-t001:** Baseline characteristics of the subjects by group (mean age, sex, weight, height, SpO_2_ and HR).

Measurement	COVID-19 Group (*n* = 15)	Non-COVID-19 Group (*n* = 15)	*p*
Mean age (years)	81.47 (5.86)	76.87 (5.08)	0.029
Sex (M/F)	6/9	6/9	-
Mean weight (kg)	69.04 (16.42)	71.49 (9.46)	0.620
Mean height (m)	1.56 (0.08)	1.61 (0.05)	0.070
Mean SpO_2_ at rest (%)	97.80 (1.08)	98.66 (0.48)	0.013
Mean HR at rest (bpm)	73.87 (8.23)	73.40 (11.94)	0.902

**Table 2 behavsci-13-00381-t002:** Mean distance covered in the 6-MWT by sex in both groups.

	Group	Female (*N*)	Mean Distance (m) (SD)
Covered distance	COVID-19	9	177.77 (41.46)
	non-COVID-19	9	250.55 (60.95)
		Mean difference	−72.77
	**Group**	**Male (*N*)**	
Covered distance	COVID-19	6	161.66 (46.22)
	non-COVID-19	6	345.83 (60.94)
		Mean difference	−184.16

**Table 3 behavsci-13-00381-t003:** The differences between groups recorded in the 6-MWT.

Group	*N*	Mean Covered Distance (m)	SD	Mean Diff.	*p*
COVID-19	15	171.33	42.57	−117.33	0.000
non-COVID-19	15	288.66	76.05		

**Table 4 behavsci-13-00381-t004:** Reliability statistics and inter-item correlations of WHOQoL-BREF Questionnaire.

Cronbach’s Alpha	0.961
Inter-Item Correlation	0.521

**Table 5 behavsci-13-00381-t005:** Initial and final scores of the WHOQoL Questionnaire.

		Initial Assessment	Std. Deviation	Final Assessment	Std. Deviation	Mean Difference	*p*	Cohen’s d
post-COVID-19 (*n* = 15)	Physical health (D1)	50.53	15.75	68.73	11.87	+18.20	0.001	1.30
	Mental health (D2)	55.53	15.61	73.06	14.22	+17.53	0.001	1.17
	Social relationships (D3)	30.80	11.50	36.20	12.85	+5.40	0.010	0.44
	Environment (D4)	47.73	12.52	58.00	12.12	+10.26	0.001	0.83

**Table 6 behavsci-13-00381-t006:** Baseline characteristics of the participants at the initial and final assessment.

Measurements	Initial Assessment	Final Assessment	Mean Difference	*p*	Cohen’s d
Mean age (SD) (years)	81.46 (5.86)	81.73 (5.66)	+0.27	0.046	-
Sex (M/F)	6/9	6/9	-	-	-
Mean weight (SD) (kg)	69.04 (16.42)	67.70 (15.88)	−1.33	0.001	-
Mean SpO_2_ at rest (SD) (%)	97.80 (1.08)	98.73 (0.45)	+0.93	0.004	1.12
Mean HR at rest (SD) (bpm)	73.86 (8.23)	70.26 (8.54)	−3.60	0.001	0.42
Distance covered in the 6-MWT (m)	171.33 (42.57)	218.00 (44.43)	+46.66	0.001	1.07

## Data Availability

The data are not publicly available due to privacy or ethical restrictions.

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
