# Peer review of "Study of Exercise Capacity and Quality of Life after SARS-CoV-2 Infection among the Elderly"

_behavsci, 2023, doi:10.3390/bs13050381_

Round 1

Reviewer 1 Report (Previous Reviewer 1)

Dear authors,

Author Response

Dear Mr./Ms.,

Thank you again for reviewing my manuscript. I appreciatte your suggestions. Please see the attachement for detailed answers.

Reviewer 2 Report (Previous Reviewer 3)

The authors have answered almost all my questions.
But there are other comments that have not been resolved and that I consider to be important.

1. The Standard Deviation (SD) is a measure of dispersion, so when it is accompanied by the ± sign it can be confusing. This ± sign is accompanied by the Standard Error (SE) since this error is a measure of precision. I suggest using Mean (SD) or Mean ± SE.

There is no table 5.

The graph should be self-explanatory and the axes are not understood. You should put on each axis what is represented.

Author Response

Dear Mr./Ms.,

Thank you again for reviewing my manuscript. I appreciatte your suggestions. Please see the attachement for detailed answers.

Reviewer 3 Report (New Reviewer)

Review

Dear authors,

I am glad I had the opportunity to review this manuscript, it is a well designed study. The results are relevant.I just make a few mentions, Please chech the order of citations, 2 is followed by 4 , line 26-32.

The conclusions should be shortened and structured in the form of a take-home message.What the conclusions now contain should be moved to discussions.

Congratulation.

Author Response

Dear Mr./Ms.,

Thank you again for reviewing my manuscript. I appreciatte your suggestions. Please see the attachement for detailed answers.

Reviewer 4 Report (New Reviewer)

The subject is interesting and should be studied. The study, I believe, should be improved according to some aspects: English, increasing the sample number, adding the limitations of the study (questionnaire), adding a flowchart or schematic design. Some information sounded at odds with the literature....for example, how to say that a group of elderly people did not have covid, without doing an antibody analysis? with these improvements it can claim future publication.

Author Response

Dear Mr./Ms.,

Thank you again for reviewing my manuscript. I appreciatte your suggestions. Please see the attachement for detailed answers.

Round 2

Reviewer 4 Report (New Reviewer)

Reject.

Author Response

Thank you!

Point 1: The study, I believe, should be improved according to some aspects: English………….

Response 1: Thank you! The manuscript was checked regarding English language during revision.

Point 2: The study, I believe, should be improved according to some aspects: increasing the sample number………..

Response 2: Thank you! In the discussion section I mentioned the small number of participants as a limitation of the research. I also explained in the Materials and methods section that the elderly care centers accommodate an approximate number of 250 patients, more than half of them being immobilized, unable to take care of themselves.

Point 3: The study, I believe, should be improved according to some aspects: adding the limitations of the study (questionnaire)…………..

Response 3: The limitations regarding the WHOQOL-BREF questionnaire are about not specifically evaluating the quality of life after COVID-19. It was not applied before and after the infection but just before and after rehabilitation program. From now on we could evaluate the persistence of the quality of life improvement by aplying the questionnaire after a longer period of time.

Point 4: Some information sounded at odds with the literature....for example, how to say that a group of elderly people did not have covid, without doing an antibody analysis?

Response 4: Thank you for your opinion!

There is literature supporting the idea that antibody levels in patients who have had a less severe experience of acute infection tend to fade rapidly. An antibody analysis does not confirm 100% the fact that the person was never infected with SARS-CoV-2.

The literature says also that asymptomatic patients are able to infect other people. This fact is extremely dangerous for the community and the transmission of the virus (especially a community of elderly) since the identification of an asymptomatic patient is impossible without performing a test.

In Romania (including the city where the study took place), weekly RT-PCR tests were performed to each patient in a first phase. After that, rapid antigen tests were distributed in places like elderly care centers, schools, child care centers, precisely in order to avoid that an infected but asymptomatic person to spread the virus. The subjects included in this study were tested weekly with RT-PCR and then twice a month with rapid antigen tests and were admitted in the study as a result of the anamnesis and based on the results of the tests registered in their medical history.

“The viral load that was detected in the asymptomatic patient was similar to that in the symptomatic patients, which suggests the transmission potential of asymptomatic or minimally symptomatic patients. These findings are in concordance with reports that transmission may occur early in the course of infection [1; 2; 3].”

“Our findings raise concern that humoral immunity against SARS-CoV-2 may not be long lasting in persons with mild illness, who compose the majority of persons with Covid-19. It is difficult to extrapolate beyond our observation period of approximately 90 days because it is likely that the decay will decelerate [4].”

  1. Chavez, S., Long, B., Koyfman, A., & Liang, S. Y. (2021). Coronavirus Disease (COVID-19): A primer for emergency physicians. The American journal of emergency medicine, 44, 220–229. https://doi.org/10.1016/j.ajem.2020.03.036
  2. Zou, L., Ruan, F., Huang, M., Liang, L., Huang, H., Hong, Z., Yu, J., Kang, M., Song, Y., Xia, J., Guo, Q., Song, T., He, J., Yen, H. L., Peiris, M., & Wu, J. (2020). SARS-CoV-2 Viral Load in Upper Respiratory Specimens of Infected Patients. The New England journal of medicine, 382(12), 1177–1179. https://doi.org/10.1056/NEJMc2001737
  3. Rothe, C., Schunk, M., Sothmann, P., Bretzel, G., Froeschl, G., Wallrauch, C., Zimmer, T., Thiel, V., Janke, C., Guggemos, W., Seilmaier, M., Drosten, C., Vollmar, P., Zwirglmaier, K., Zange, S., Wölfel, R., & Hoelscher, M. (2020). Transmission of 2019-nCoV Infection from an Asymptomatic Contact in Germany. The New England journal of medicine, 382(10), 970–971. https://doi.org/10.1056/NEJMc2001468
  4. Ibarrondo, F. J., Fulcher, J. A., Goodman-Meza, D., Elliott, J., Hofmann, C., Hausner, M. A., Ferbas, K. G., Tobin, N. H., Aldrovandi, G. M., & Yang, O. O. (2020). Rapid Decay of Anti-SARS-CoV-2 Antibodies in Persons with Mild Covid-19. The New England journal of medicine, 383(11), 1085–1087. https://doi.org/10.1056/NEJMc2025179

This manuscript is a resubmission of an earlier submission. The following is a list of the peer review reports and author responses from that submission.

Round 1

Reviewer 1 Report

Thank you to give me the opportunity to review the manuscript. It is interesting, but some concerns should be highlighted.

Abstract is incomplete and does not meet the instructions of authors. It must contain enough information on method, results and conclusion. The authors finished the abstract with the aim. Please rewrite this section.

INTRODUCTION

2nd paragraph - The authos stated "A lot of authors have tried to quantify the impact of the persistent symptoms of COVID-19 on physical function, cognitive function, health-related quality of life and participation in social activities." However, any evidence is shown. Please presente the body of evidence supporting the statement.

The same concern here: Immune system disruption triggered by infection could induce psychopathology, with evidence of psychiatric sequelae following previous coronavirus outbreaks. It was concluded that the spread of the SARS-CoV-2 pandemic associates important psychological implications.

4th paragraph - Besides this work is part of a PhD thesis (congratulations!), I suggest a modification in this sentence. It is unusual. I recommend the following: The research topic addressed in this article represents a subject of global interest, considering the spreading of the virus and the extent of deaths caused by COVID-19.

The authors mentioned the 6MWT and WHOQoL  in the end of the introduction, explaining these instruments were used to evaluate aerobic capacity and quality of life, respectively. It is unecessary. Instead, they must explain them in a detailed fashion just in the method.

MATERIALS AND METHODS

Although the authors mentioned the ethical compliance in the end of the manuscript, I recommend to insert this information in this section adding the approval protocol number

Sample size - Sample is too small and it compromises the power of the study. How many older adults lived in each long-term facility? It could help authors to justify the sample size.

Please write a complete name of the WHOQoL-BREF at the first time it appears in the text.

Please provide basic information about the interpretation of the results (and classification) from 6MWT.

Statistical analysis must appear in the end of the method. Please inser the explanations regard WHOQoL in a topic immediately after the 6MWT.

As the sample size is too small, I am afraid the normal distribution and homocedasticity were not achieved. Did the authors check these assumptions? Please insert a sentence mentioning wether dependent variables reached the aforementioned assumptions and which tests were used. If the assumptions were violated, Mann-Whitney U test should be applied instead Independent t-tes.

There was no analysis to check differences between male and females in each group. I suggest to apply the Fisher exact test.

"Assesments" is a typo.

Authors stated " Pearson Correlation was used in order to confirm or exclude the fact that age of the participants in this specific study had an influence on the results obtained at the 6-Minute walk test". However, this analysis cannot be used to predict cause and effect (as influence). Instead, it is a simple correlation where one variable presents its data positively/negatively proportional (or not) to another. I recommend the authors rewrite this sentence.

RESULTS

1st paragraph is not really a result, rather it is an interesting comment based on the literature. I suggest to replace this sentence in the discussion.

An important pitfall is the significant difference of age between groups. As the Covid-group was older than non-Covid one, it could influence the results of aerobic capacity. Literature is quite aligned that aerobic capacity (as other physical capacities) is inversely proportional to age. The question is: The worse performance of Covid-group in the 6MWT is due the infection or age (or both)? Besides the authors have tried to minimize it, Pearson´s correlation does not provide substantial information to confirm the author´ statement in the 7th paragraph of the results (this paragraph must be completely re-writen. Otherwise, it enhances the doubt. An ideal analysis would be regression with age as covariate. However, it is not possible with small sample size. I believe the findings are going on a correct direction, but it cannot be confirmed herein. It is a relevant limitation and must be stated in the end of the discussion.

P-value .000 is unusual. Please rewrite as <.01.

Authors did not mention any intervention program in the method. It sounds like misorganized writing. I really recommend insert a subtopic about this in the method.

Supposed results from the rehabilitation program should be interpreted carefully because there is no control group. Please moderate their statements (i.e. "suggests that differences recordered between mean weight, mean SpO2 at rest, HR and distance covered at 6-MWT are statistically significant between the 2 assesments, meaning that the implementation of a rehabilitation program improved not only quality of life butalso the effort capacity among elderly subjects who went through COVID-19"). People who receive care and engage in social activities improve mental health and social relationships. Authors must point out the absence of control group as limitation.

According to the American College of Sports Medicine and other Exercise Physiology guidelines, 60% HRmax is light to slightly moderate intensity instead high intensity. Please  verify and adjust adequately. 

Table 7 shows a small difference in the age of the group between initial and final assessments. It means there was drop-out in the sample. Please clarify.

SpO2 increased in the final assessment compared to initial assessment, but the mean difference is write as negative. Please verify.

Something similar with HR and 6MWT, but in the opposite direction.

Discussion and conclusion must be re-written considering the aforementioned alterations, since the results may change.

FINAL CONSIDERATIONS

Manuscript has many flaws and must be carefully reviewed and resubmitted. I recommend a resubmission as short communication or even as case series (type of article case report).

Author Response

Dear Reviewer,

First of all I want to thank you for your review report.
It helped me to see more clearly the results and outcomes I obtained.
I have made the corrections you`ve suggested. Please see the attachment.

Thankfully,
Vrabie Diana

Reviewer 2 Report

Dear Authors,

this is an interesting article examining effort capacity and quality of life after SARS-CoV-2 infection. I think that this analysis is very interesting but needs some corrections. The statistical applications in the second part of the work are not appropriate.

The statistical analysis for the sample of 15 subjects, shown in tables number 6 and 7, should use a non-parametric test due to the small number of the sample (<30 subjects). Therefore It should be applied a test for paired samples since it is the same sample analyzed in two times (before and after). The Wilcoxon paired-sample test should be used. This could bring variations in the results and interpretation of the data into the discussion.

Minor revision:

Results Page 5 – Line 169. There should be an error: it is not “minute 4 after start” but  “3'30" after start” (verify the reported data).

Author Response

Dear Reviewer,

Thank you for your review report. It helped me to see more clearly the outcomes I obtained.

I have made all the corrections you have suggested. Please see the attachment.

Thankfully, 

Vrabie Diana

Reviewer 3 Report

Dear authors,

The paper is interesting as it deals with a current issue and the quality of life of elderly patients.

However, I have some comments to make:

Major Issue

There is no indication of whether a study of the sample size and statistical power has been performed in the case of 15 participants per study group.

The criteria for inclusion and exclusion of participants in the study are not indicated.

I believe that, due to the characteristics of the sample, nonparametric tests should be used instead of Student's t-test. The t-tests need several conditions to be applied and these variables do not meet them.

In the Results section, the type of analysis performed is indicated again when it is already indicated in the Data analysis section.

The difference between the means is indicated as results. This difference must be relativized by the standard deviations. I suggest showing the Coefficient of Variation or measuring the size of the effect with Cohen's d.

In Table 3 an assessment of the results is made and this assessment should be in the Discussion section.

Table 3 (bis) discusses the limitation of calculating the Pearson Correlation Coefficient. I suggest calculating the Spearman Correlation Coefficient.

Do the same in Table 4.

The graph should be self-explanatory and the axes are not understood. You should put on each axis what is represented.

The results, although I assume they are true, may not be correct.

Therefore, the conclusions may not be correct.

Minor issue

Line 42 talks about the completion of a thesis and I think it should be removed from the text.

There are two Tables 3.

In Table 1: in the text it says PS02 instead of SpO2.

The Standard Deviation (SD) is a measure of dispersion, so when it is accompanied by the ± sign it can be confusing. This ± sign is accompanied by the Standard Error (SE) since this error is a measure of precision. I suggest using Mean (SD) or Mean ± SE.

- Altman DG, Gore SM, Gadner MJ, Pocock SJ. Statistical guidelines for contributors to medical journals. Br Med J 1983;286: 1,489-1,493.

- Bailar JC, Mosteller F. Guidelines for statistical reporting in articles for medical journals: amplifications and explanations. Ann Intern Med 1988;108: 266-273.

- Tobias A. [Mean +/- SD, an incorrect expression].Med Clin (Barc). 1998 Feb 7;110(4):157

Author Response

(The authors gave the same response as above.)
